# Orthogonally-tunable and ER-targeting fluorophores detect avian influenza virus early infection

Taewon Kang[1,4], Md Mamunul Haque [2,4], Boran Lee[1,4], Kyung Tae Hong[3], Seong Cheol Hong[3], Younghun Kim [1], Jesang Lee[1], Jun-Seok Lee [2] ✉ & Dongwhan Lee [1] ✉

Cell-based assays can monitor virus infection at a single-cell level with high sensitivity and cost-efficiency. For this purpose, it is crucial to develop molecular probes that respond selectively to physiological changes in live cells. We report stimuli-responsive light-emitters built on a T-shaped benzimidazole platform, and consecutive borylation reactions to produce a library of homologs displaying systematic changes in fluorescence quantum yield and environmental sensitivity. We find that certain fluorophores localize selectively at the endoplasmic reticulum, and interact with proteins involved in the stress signaling pathways. Notably, the mono-borylated compound responds selectively to the stress conditions by enhancing fluorescence, and detects avian influenza virus infection at the single-cell level. Our findings demonstrate the unprecedented practical utility of the stress-responsive molecular probes to differentiate cellular states for early diagnosis.

Fluorescent molecules are at the forefront of chemical tools to detect subtle changes in cells and subcellular organelles. With appropriate structure design, such optical probes can respond to spatiotemporal changes in the cellular environments, so that biologically important information can be obtained by tracking shifts in the emission energy, intensity, or a combination of both[1–5]. For example, simultaneous use of multiple fluorescent probes, each responding sensitively and selectively to different types of physiological parameters, could be useful for an early diagnosis of disease. In particular, modern high contents screening (HCS) studies have demonstrated the usefulness of chemical probes for biological functional study, toxicology, and screening of bioactive molecules[6–8].

Despite significant advances, however, there still remain great demands for functional probes that can discriminate specific cellular states, in which a single biomarker alone cannot perform perfectly. For instance, the differentiation of virus infections is a complex biological process that is orchestrated by many transcriptional and translational controls. As such, markers for cellular states can be detected by antibody-based method only after lysis or under fixation conditions. Within this context, environmentally sensitive and live-cell compatible fluorescent probes that discriminate cellular states have significant scientific implications and practical impacts. For biological applications, fluorogenic molecules need to have bright emissions for high signal-to-noise ratio imaging, optical responsiveness to environmental changes, and low cytotoxicity for in vivo studies as well. The fluorogenic benzimidazole platform[9–11] could be one such candidate. From a structure–properties point of view, structural rigidification of the π-conjugation would enhance the fluorescence quantum yield[12,13], but the removal of hydrogen bond donor and acceptor (HBD–HBA) motifs might effectively shut down environment-sensitive emissive pathways such as excited-state intramolecular proton transfer (ESIPT)[14–16].

In this paper, we report the chemistry of T-shaped fluorogenic molecules EliF (= **E**xtended and **L**igating **I**midazolyl **F**luorophore, Fig. 1). Key features of the EliF molecules include (i) high modularity in synthesis to facilitate systematic variation of the electronic properties of the π-skeleton, and (ii) straightforward main-group coordination

[1]Department of Chemistry, Seoul National University, 1 Gwanak-ro, Gwanak-gu, Seoul 08826, Korea. [2]Department of Pharmacology, Korea University College of Medicine, 73 Goryeodae-ro, Seongbuk-gu, Seoul 02841, Korea. [3]Bio-Med Program, KIST-School UST, Hwarang-ro 14 gil 5, Seongbuk-gu, Seoul 02792, Korea. [4]These authors contributed equally: Taewon Kang, Md Mamunul Haque, Boran Lee. ✉e-mail: junseoklee@korea.ac.kr; dongwhan@snu.ac.kr

chemistry to convert the free ligands to the mono- and bis(BF₂)-adducts having distinctively different photophysical properties. Even without installing organelle-targeting functional groups, certain EliF derivatives are taken up by live cells to become localized selectively at the endoplasmic reticulum (ER). Moreover, an ER localizing probe responds to early-stage of avian influenza (AI) virus infection by recognizing upregulated proteins in the ER stress signaling pathway of live cells.

## Results and discussion

### Construction of orthogonal π-conjugation: oxidative cyclization and borylation

As shown in Fig. 1, The **EliF-1** series molecules are built by appending two pyridyl groups to the central *N*-aryl benzimidazole unit. In addition to laterally extending the π-conjugation to define the overall T-shape, these pyridyl groups take part in a contiguous array of N–H···N and C–H···N hydrogen bonds to help planarize the π-conjugation. We anticipated that structural changes here, either (i) by modulating the strength of hydrogen bonds by local dielectric or (ii) by more drastic covalent modification such as borylation of the *N*-donor groups, could profoundly impact the photophysical properties of EliF. To realize these ideas with actual molecules, a modular synthetic strategy was devised. As outlined in Fig. 2, the diamine compound **2**, derived from the benzothiadiazole precursor **1**, was employed as a common intermediate, and subjected to oxidative condensation reactions with various aryl aldehydes. With **EliF-1a** having a 4-ethylphenyl substituent as a reference system, 4-cyanophenyl (**EliF-1b**), 4-dimethylaminophenyl (**EliF-1c**), and 2-pyridyl (**EliF-1d**) groups were installed as electron-withdrawing, electron-donating, and heteroatom-containing aryl substituent, respectively, onto the common benzimidazole core.

By design, the two [*N,N*]-bidentate pockets of **EliF-1** could function as donor groups for the coordination of electrophilic anchoring

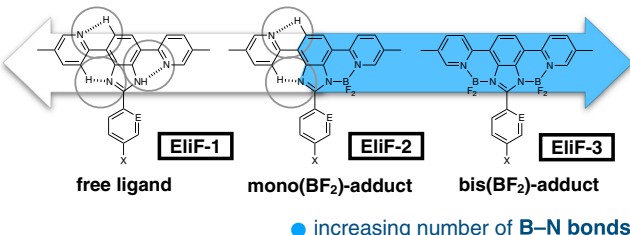

- increasing number of **hydrogen bonds**
- preorganized yet **flexible**
- more **responsive**

**EliF-1** — free ligand
**EliF-2** — mono(BF₂)-adduct
**EliF-3** — bis(BF₂)-adduct

- increasing number of **B–N bonds**
- conformationally **rigid**
- more **tolerant**

**Fig. 1 | Design principles of EliF.** Main-group coordination chemistry converts non-covalent N–H···N hydrogen-bonding contacts to rigid B–N coordinate bonds, which profoundly impacts the environment-sensitivity and emission quantum yield of the fluorogenic skeleton.

**Fig. 2 | Modular construction of EliF molecules. a** Synthetic routes to free ligands and stepwise borylation. **b** Representative single-crystal X-ray structures. Et ethyl, DMF *N,N*-dimethylformamide, Me methyl, DIPEA *N,N*-diisopropylethylamine.

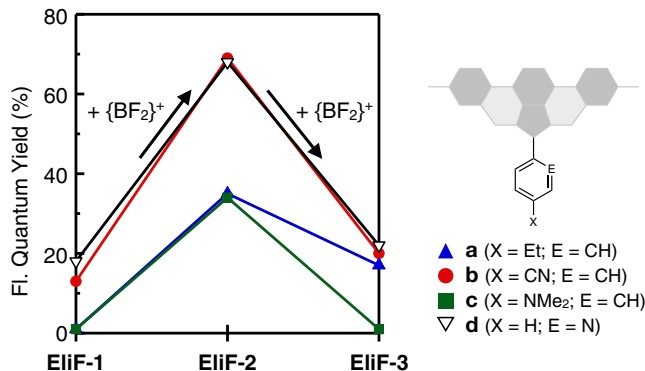

**Fig. 3 | Structure-dependent light-emitting properties.** Changes in the fluorescence emission quantum yield ($\Phi_F$) of **EliF-a** (blue line), **EliF-b** (red line), **EliF-c** (green line), and **EliF-d** (black line) series molecules in CHCl₃ ($T = 293$ K; $\lambda_{exc} =$ 350 nm) with progressive installation of {BF₂}⁺ fragments onto the [$N,N$]-bidentate pocket. Fl fluorescence.

groups, such as {BF₂}⁺, to serve as proton surrogates. Through variations in the choice of solvent, temperature, and reaction time, we were able to convert selectively each of the **EliF-1** free ligands into the corresponding mono(BF₂)-adducts **EliF-2**, and bis(BF₂)-adducts **EliF-3** (Fig. 2). A total of 12 compounds having different substitution patterns and overall charges were thus prepared in a straightforward fashion, by taking full advantage of the high modularity in fragment assembly and minimal synthetic operations. In large part, the success of our synthesis benefits from the differential reactivity of the two $N$-donor groups, and the overall charge of the boron adducts. After installation of the first {BF₂}⁺ unit at the anionic and more reactive [$N,N$]-pocket of the deprotonated **EliF-1** to afford charge-neutral **EliF-2**, the second borylation step occurs at the remaining charge-neutral (and thus less reactive) [$N,N$]-pocket to produce cationic **EliF-3** (Figs. 1 and 2). During the course of this stepwise transformation, the cationic bis(BF₂)-adducts form ionic salts with the BF₄⁻ anion generated in situ, and precipitate out of the reaction mixture. A simple filtration and washing cleanly afforded analytically pure bis(BF₂)-adducts, without further purification by anion metathesis or recrystallization.

As shown in Fig. 2 and Supplementary Figs. 1–7, the single-crystal X-ray structures unambiguously establish the presence of multiple intramolecular hydrogen bonds in the free ligands, either N–H···N or C–H···N. Upon coordination of an increasing number of {BF₂}⁺ fragments, the number of HBD–HBA contacts decreases systematically. This trend correlates nicely with the decrease in the number of rotatable C–C bonds along with the series **EliF-1** → **EliF-2** → **EliF-3** (Fig. 1), with the coordinated {BF₂}⁺ functioning as a conformational lock. We anticipated that this structural rigidification would produce brighter light-emitters, whereas replacing the hydrogen bonds could attenuate environmental sensitivity. To probe the photophysical consequences of this tradeoff model (Fig. 1), we proceeded to carry out comparative photophysical studies.

### Structure-dependent and environment-sensitive fluorescence properties

As summarized in Fig. 3 and Supplementary Table 1, installation of the first {BF₂}⁺ unit (i.e., **EliF-1** → **EliF-2**) dramatically enhances the emission quantum yields by as much as 70%. At the same time, the solvatochromic response of the fluorophore is diminished significantly as a tradeoff (Fig. 4 and Supplementary Figs. 8–10). The most noticeable change is a complete disappearance of the dual emission (Fig. 4a and Supplementary Fig. 8a) from ESIPT. For **EliF-1** having N–H···N hydrogen bond, the ESIPT reaction is under kinetic control (Supplementary Fig. 12), allowing both tautomers to take radiative pathways to produce dual emission. This theoretical prediction is supported by a relatively

short fluorescence lifetime of the local emission and a double-exponential fitted decay of the tautomer emission of **EliF-1b** (Supplementary Table 1). The strikingly different photophysical properties of **EliF-1** vs. **EliF-2** series are consistent with the anticipated structure–property correlation in Fig. 1. Replacement of the N–H···N hydrogen bonds with rigid B–N coordinate bonds simplifies both the conformational space and excited-state structural dynamics to produce a brighter emitter, but lowers its sensitivity to the environment.

Upon installation of the second {BF₂}⁺ unit (i.e., **EliF-2** → **EliF-3**), however, a rather counterintuitive behavior was observed. While the solvatochromic response of the bis(BF₂)-adducts is comparable to that of mono(BF₂)-adducts (Supplementary Figs. 8–10), the fluorescence quantum yields become significantly lowered (Fig. 3 and Supplementary Fig. 11). Apparently, overborylation of **EliF-2** to **EliF-3** not only rigidifies the structure, as reflected on the evolution of well-resolved vibronic features in the emission spectra (Fig. 4 and Supplementary Figs. 8–11), but also opens up undesired non-radiative channels that diminish the fluorescence efficiency, as reflected on the increased $k_{nr}$ values determined by lifetime measurements (Supplementary Table 1). As a result, **EliF-3** becomes less responsive to the environment than the free ligand **EliF-1**, and less emissive than the mono(BF₂)-adduct **EliF-2**.

To gain a detailed understanding of the structure-dependent changes in the light-emitting properties (Figs. 3 and 4), we carried out time-dependent density functional theory (TD-DFT) computational studies (Supplementary Figs. 13–16). A close inspection of the shapes of the FMOs suggests that the electronic excitation of **EliF-3** has a significant vertical-to-horizontal intramolecular charge-transfer (ICT) character (Supplementary Figs. 13 and 16). The absence of bathochromic shifts in polar solvents (Fig. 4c, and Supplementary Figs. 8c, 9c, and 10) implicates that the charge-transfer leading to a dark ¹ICT state is responsible for the diminished emission quantum yield of **EliF-3** (Fig. 3). By suppressing this deleterious ICT pathway, electron-withdrawing aryl substituents in the vertical π-conjugation tend to produce brighter emission, as supported by the plot of fluorescence quantum yields vs Hammett constants $\sigma_p$ (Supplementary Fig. 17)[17,18].

With a better understanding of the photophysics of EliF molecules, we proceeded to test their practical utility. It was anticipated that the less-bright-yet-more-responsive **EliF-1** (Figs. 3 and 4a) could be useful as fluorescence sensors, whereas bright-and-robust **EliF-2** (Figs. 3 and 4b) as bio-imaging agents. As detailed in the following sections, our exploratory studies have revealed that certain EliF molecules selectively target the endoplasmic reticulum (ER).

### Fluorescence live-cell imaging and discovery of endoplasmic reticulum (ER) localization

To evaluate the practical application of EliF molecules for biological imaging, we initially checked the toxicity of each compound using a luminescence-based cell viability assay (Supplementary Fig. 18). None of the compounds exhibit significant dose-dependent toxicity at the micromolar level. While the **EliF-1** series molecules become more toxic under UV irradiation ($\lambda = 350$ nm) conditions, photo-toxicity diminishes with increasing level of borylation: **EliF-1** ≫ **EliF-2** > **EliF-3** (Supplementary Fig. 19). Since the photophysical properties of the EliF system are yet to be optimized for conventional fluorescence microscopy, we customized the excitation/emission filters for optimum signal acquisition by incubating each compound in HeLa cells and monitoring live-cell staining patterns (Supplementary Fig. 20).

Among the twelve EliF compounds listed in Fig. 2, three molecules (**EliF-1a**, **1c**, and **3c**) showed no or very dim fluorescence staining, possibly due to their low quantum yield and/or low cell permeability. On the other hand, the remaining nine compounds were successfully permeabilized and stained the cytosolic part of HeLa cells. Since the EliF scaffold itself is charge-neutral and contains multiple aromatic units, we initially anticipated that their cellular accumulation pattern might be similar to that of BODIPY-like fluorophores, which

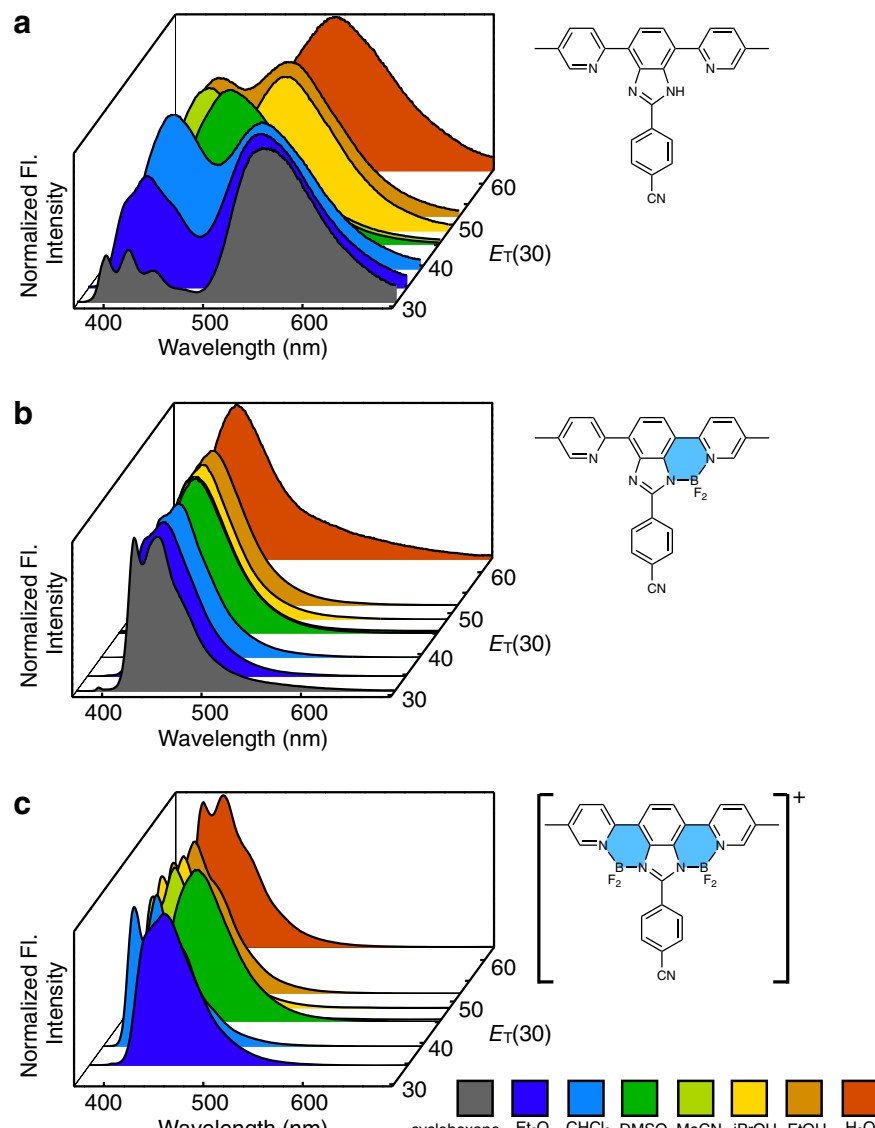

**Fig. 4 | Solvent-dependent changes in the fluorescence spectra.** Normalized fluorescence spectra of **EliF-1b** (**a**), **EliF-2b** (**b**), and **EliF-3b** (**c**) in cyclohexane (gray), Et$_2$O (blue), CHCl$_3$ (sky blue), DMSO (green), MeCN (yellow-green), *i*PrOH (yellow), EtOH (orange), and water (red) from front to back corresponding to increasing solvent polarity as denoted by the $E_T(30)$ values[19] in the *y*-axis ($T = 293$ K; $\lambda_{exc} = 350$ nm). DMSO dimethyl sulfoxide, *i*Pr isopropyl.

tend to have high staining signals and localize at hydrophobic membranes[20,21]. Unlike our expectation, however, EliF series molecules exhibited dimmer signals under the microscope but unique staining with larger Stokes shifts. Moreover, similar staining patterns were observed despite the structural distinctions of each compound. The fluorescence cell images revealed interconnected network sacs and tube-like structures throughout the individual cell's cytosol region (Fig. 5). Based on the stained structural characteristics, we suspected that EliF was accumulated in subcellular organelles.

To verify which organelle was labeled by the EliF dyes, we conducted a co-localization experiment using a series of organelle trackers. Compounds that show fluorescence signals from all microscope filter sets could not be examined by a co-staining method. We thus ruled out four such compounds (**EliF-2a**, **2b**, **3a**, and **3b**), and focused on the remaining five (**EliF-1b**, **1d**, **2c**, **2d**, and **3d**). To visualize organelles, HeLa cells were transfected with each CellLight™ organelle tracker (Endoplasmic Reticulum (ER)-RFP, Mitochondria-RFP, Golgi-RFP, Lysosomes-RFP, Peroxisome-GFP, and NUCLEAR-ID Red) for 24 h to express fluorescent proteins, and the

EliF dyes were administrated into the cells for imaging (Supplementary Figs. 21–25).

Our co-localization images unambiguously established that the EliF dyes are localized in the ER. ER is a subcellular organelle that plays crucial roles in protein glycosylation, folding, lipid biosynthesis, and calcium homeostasis[22–24]. As such, ER dysfunction is related to multiple diseases, including type 2 diabetes, neurodegenerative disease, ischemia, and virus infections[25,26]. Due to the difficulty of controlling ER selectivity, few organic fluorescent probes are currently available that can target ER[27]. These conventional probes are categorized into two classes. One class is glibenclamide conjugated fluorophores that are designed to bind to an ER protein, ATP-sensitive potassium channel[28–30]. The other class has been empirically discovered, such as 2,5-diphenyloxazoles derivative that was dubbed ER blue-white DPX[31]. These organic dye trackers have practical merits, such as short incubation time (24 h for expression of fluorescent protein (FP) vs 30 min for cellular distribution of dye) and ease of dose control, compared with FP expression-based CellLight™ organelle trackers. However, their organelle selectivity and photostability differ depending on the

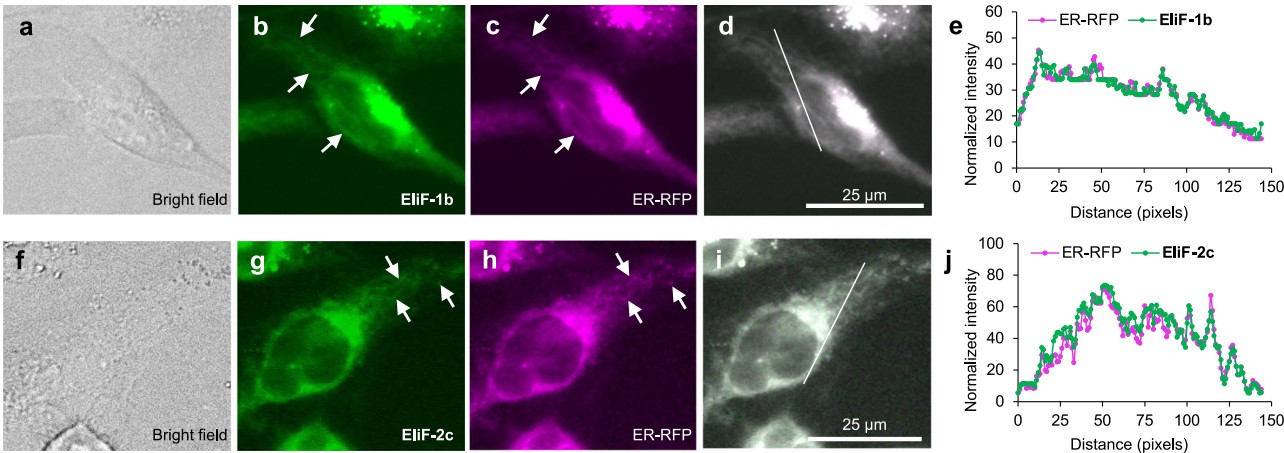

**Fig. 5 | Fluorescence co-localization images of live HeLa cells stained with ER-RFP and EliF probes.** Panels **a**–**e** correspond to **EliF-1b**; **f**–**j** to **EliF-2c** (sample concentration = 10 μM; cell incubation time = 30 min). Panels **b**, **g**: fluorescence images of EliF probes obtained using F1 filter-set ($\lambda_{exc}$ = 350/50 nm, $\lambda_{em}$ = 527/30 nm). Panels **c**, **h**: fluorescence images of ER-RFP obtained using RHOD filter-set ($\lambda_{exc}$ = 546/10 nm, $\lambda_{em}$ = 585/40 nm). In panels, **b**, **c**, **g**, **h**, white arrows mark sac or tube-like structures of ER. Panel **d** is the merged image of **b** and **c**; panel **i** is the merged image of **g** and **h**. Co-localization patterns shown in **e** and **j** are plotted as fluorescence intensity profiles of ER-RFP (magenta line) and EliF probes (green line) in the regions of interest, which are indicated by white lines in the images **d** and **i**, respectively (scale bar = 25 μm). Imaging experiments were conducted on at least three independent biological replicates; all replicates produced similar results.

structure of the ligands and fluorophores. As shown in Fig. 5, EliF clearly visualizes ER network structures. As the EliF probes showed prominent ER localization, we further compared their photostability with a commercial organic dye tracker, ER-Tracker™ Green. Both **EliF-1b** and **EliF-2c** showed comparable stability with the commercial tracker (Supplementary Fig. 26).

### Cellular targets of EliF probes

While fluorescence co-localization studies suggested ER localization, it was not straightforward to understand the sub-organelle specificity of the EliF probes. We thus proceeded to identify the cellular targets of **EliF-1b** and **EliF-2c**. As the EliF molecular scaffold has not been investigated for cellular target proteins, unbiased target engagement profiling was needed. For this purpose, several identification strategies are currently available, such as affinity matrix, in situ labeling, and label-free methods[32,33]. We chose a label-free target identification using thermal proteome profiling (TPP)[34,35], since it does not require structural modification of the probe molecules that could potentially alter its cellular localization.

A given composition and sequence of protein has its unique thermal tolerance. In other words, each protein has a distinctive melting temperature. Ligand binding alters such property, and it is possible to deduce ligand–protein interactions based on melting temperature shift. Using multiplexed tandem mass-tag (TMT), TPP measures changes in the melting temperatures of proteins in cellular conditions. TPP experiments were conducted according to the previously reported procedure[36]. Briefly, HeLa cells were preincubated with **EliF-1b**, **EliF-2c**, or vehicle, and each sample was divided into ten fractions. A heat shock condition from 37 to 67 °C was applied to each fraction to induce thermal denaturing. Non-denatured proteins were subsequently recovered, digested into peptides, and labeled with 10-plex TMT. Since TMTs have identical chemical structures but differ in mass by substitution of $^{13}$C and $^{15}$N isotopes, it is possible to compare the thermal stability of each protein quantitatively.

From TPP experiments using nanoLC coupled to Orbitrap Eclipse Tribrid MS with collision-induced dissociation (CID) and higher energy CID (HCD), we identified 5512, 5245, and 6352 proteins from the cells treated with vehicle, **EliF-1b**, and **EliF-2c**, respectively (Supplementary Data 1). After applying filters based on distinctive $T_m$ values, 29 and 36 protein hits were discovered by **EliF-1b** and **EliF-2c**, respectively, with ten common hits from both compounds (Fig. 6a). The average $T_m$

difference of the primary hit proteins were 3.5 °C (Fig. 6b), and there are four proteins (PSPH, PDXDC2P, PDXDC1, and NCAPG) that exhibited $T_m$ shift larger than 7 °C. Among these four proteins, PDXDC2P and PDXDC1 are previously known for their cellular localization in ER. Further analysis indicated that 64% of the 55 primary hits are either found in ER, or involved in ER-related signaling pathways (Fig. 6c and Supplementary Data 1).

We next examined the hit proteins by bioinformatic methods that interpret the functional profile of the enriched gene sets using Gene Ontology (GO) analysis. The GO analysis (Fig. 6d and Supplementary Data 1) also revealed that the top three cellular component keywords for our hit proteins are related to the terms cytosolic and membrane-bound organelle with exceptionally high statistical significance. A combination of chemical proteomics and gene annotation analysis thus establishes that the primary target proteins of EliF probes are indeed enriched in ER, which is consistent with their selective localization confirmed by fluorescence imaging (Fig. 5).

It is noteworthy that 15 out of 55 primary hits (seven out of ten common hits) proteins are related to ER stress signaling (Supplementary Data 1). This observation led to our subsequent investigation of potential bioassay development with **EliF-1b** and **EliF-2c** for ER stress monitoring in live cells. Since many diseases or pathological conditions of acute viral infections are highly related to ER stress leading to the accumulation of misfolded proteins, there exists a high demand for simple experimental protocols to measure ER stress.

### EliF-1b and EliF-2c as ER stress-responsive probes

Several research groups have recently reported ER stress assays using luciferase reporter[37], qPCR[38], and transcription activation[39]. A few molecular probes are also available that can detect environmental changes of ER by fluorescence imaging[40–42]. Since image-based quantification methods are inherently prone to the choice of analysis parameters and algorithms employed, their response was subtle and only qualitative. In this regard, there are no molecular probes to monitor ER stress quantitatively. We envisioned that EliF has the potential to discriminate ER status because overloading of misfolded protein accumulation could lead to micro-environmental changes resembling solvatochromic effects (Fig. 4 and Supplementary Figs. 8–10).

To evaluate the fluorescence response of EliF upon ER stress, we designed a model condition using two ER stress-inducing agents,

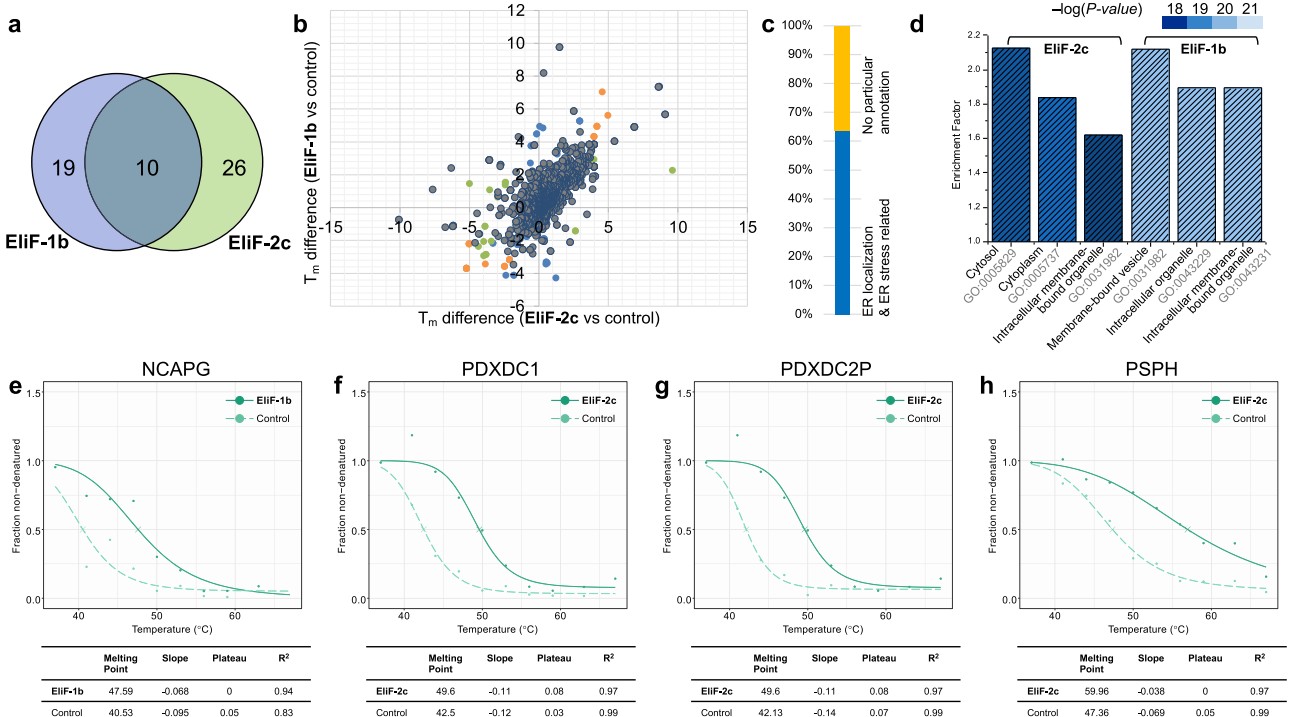

**Fig. 6 | Label-free target identification for EliF-1b and EliF-2c. a** A Venn diagram of target proteins identified by TPP. **b** $T_m$ shift plot of TPP with target proteins of **EliF-1b** in blue, **EliF-2c** in green, and the common hits in orange color, respectively. **c** ER-related functional annotation of target proteins. **d** Gene ontology enrichment analysis of 55 target proteins using the cellular component term. Thermal protein melting curves of four major target proteins, NCAPG (**e**); PDXDC1 (**f**); PDXDC2P (**g**); PSPH (**h**) treated with EliF probes (green solid line) and control groups (green dashed line).

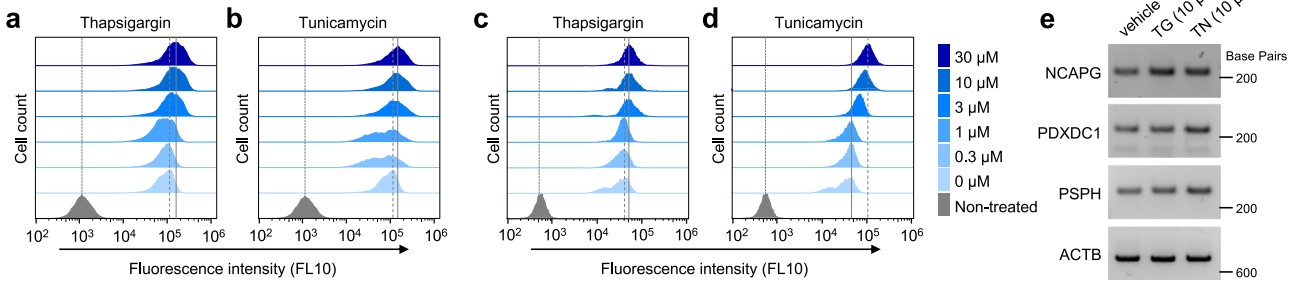

**Fig. 7 | ER stress responses.** Flow cytometry analysis of **a**, **b** EliF-1b (10 µM), and **c**, **d** EliF-2c (10 µM) in ER stress-induced HeLa cells. ER stress was chemically induced by increasing concentrations (0–30 µM) of either thapsigargin (TG; for **a** and **c**) or tunicamycin (TN; for **b** and **d**) preincubation for 12 h, and monitored by cell population shifts in flow cytometry using FL10 filter ($\lambda_{exc} = 405$ nm; $\lambda_{em} = 550/$ 40 nm). Based on the mean fluorescence intensity, **EliF-1b** showed 1.55- and 1.45-fold increment with TG and TN, respectively (concentration = 30 µM). **EliF-2c** showed 1.79- and 3.31-fold increment with TG and TN, respectively (concentration = 30 µM). **e** RT-PCR analysis of major target proteins (NCAPG, PDXDC1, and PSPH) upon ER stress induction. Source data are provided as a Source Data file.

thapsigargin and tunicamycin, which operate by distinct mechanisms. Thapsigargin (TG) causes ER stress by inhibiting ER Ca²⁺-ATPase activity[43,44], whereas tunicamycin (TN) inhibits *N*-linked glycosylation of nascent proteins, thus leading to an accumulation of hypoglycosylated proteins in the ER lumen[45,46]. HeLa cells were preincubated with varying concentrations of TG and TN (0–30 µM) for 12 h to induce ER stress, and subsequently treated with EliF molecules. To obtain quantitative fluorescence response profiles at the individual cell level, we conducted flow cytometry-based screening (Fig. 7 and Supplementary Fig. 27).

As shown in Fig. 7, both **EliF-1b** and **EliF-2c** display dose-dependent fluorescence responses upon ER stress. Intriguingly, both TG and TN induced a significant increase in the high fluorescence population of the cells stained with **EliF-1b** and **EliF-2c**. In particular,

**EliF-2c** exhibited a remarkable fluorescence increase for the TN treatment. The major cell population was shifted prominently to higher fluorescence at higher concentrations (> 3 µM) of ER stress-inducing agents, and the mean fluorescence intensity increased up to 3.31-fold for the treatment of TN (30 µM). Such response was apparently distinctive compared with other EliF compounds (**EliF-1d**, **EliF-2d**, and **EliF-3d**; Supplementary Fig. 27) or commercial ER trackers (ER-Tracker™ Blue-White, and ER-Tracker™ Green; Supplementary Fig. 28), which do not show noticeable shifts in fluorescence intensity under identical conditions. Additionally, major target proteins (NCAPG, PDXDC1, and PSPH) that exhibited significant $T_m$ shift upon binding to **EliF-1b** or **EliF-2c** were upregulated under ER stress conditions (Fig. 7e). The results from these control experiments implicate that the fluorescence responses of **EliF-1b** and **EliF-2c**

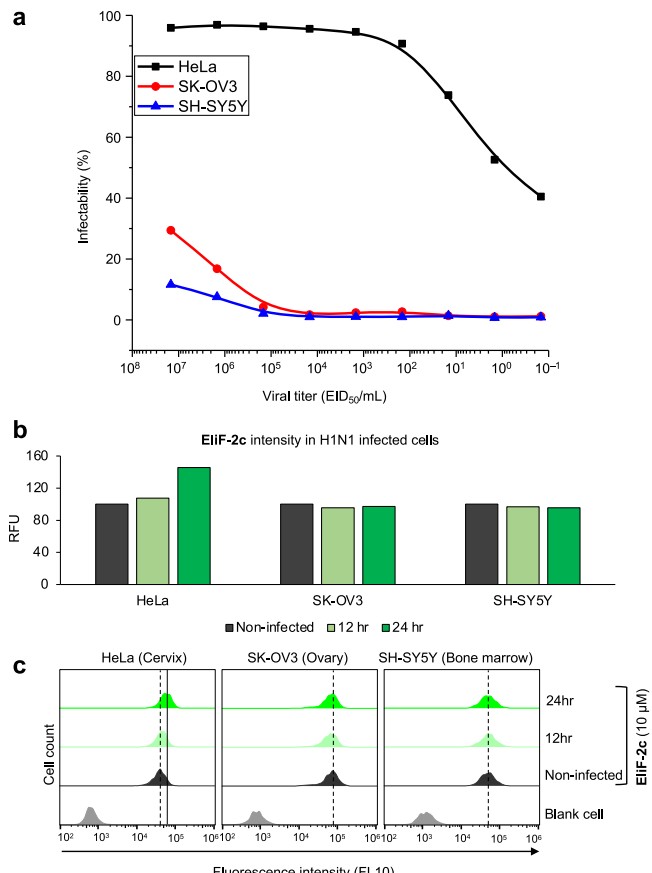

**Fig. 8 | Response to H1N1 AI infection. a** H1N1 virus infectability in HeLa (black), SK-OV3 (red), and SH-SY5Y (blue) cells after 24 h. **b** Bar graphs showing the mean fluorescence intensity of **EliF-2c** with 1.46-fold enhancement in the HeLa cell after 24 h infection. **a**, **b** Data were representative of two replicate experiments. **c** Flow cytometry analysis of **EliF-2c** (10 μM) in HeLa, SK-OV3, and SH-SY5Y cells using FL10 filter ($\lambda_{exc} = 405$ nm; $\lambda_{em} = 550/40$ nm). RFU relative fluorescence units.

(Fig. 7) do not originate from toxicity-induced non-specific cellular uptake.

We also examined whether **EliF-1b** and **EliF-2c** respond to other organelle perturbation conditions. Carbonyl cyanide *m*-chlorophenyl hydrazone (CCCP) is a well-known agent for mitochondrial membrane potential depolarization[47,48]. Neither **EliF-1b** nor **EliF-2c** showed fluorescence response to CCCP in flow cytometry experiments (Supplementary Fig. 29). These observations support that **EliF-1b** and **EliF-2c** respond selectively to ER stress signals.

### Early diagnosis of avian influenza infection by EliF probes
During the last decades, several global pandemic emergences of RNA viruses have been reported, including the 2009 H1N1 influenza and 2019 SARS-CoV-2 virus. Many techniques are currently available to diagnose RNA virus infections, but continuous infection monitoring at the individual cell level remains challenging. Our group has been investigating the host cell response to influenza virus infection using fluorescent probes and demonstrated successful discrimination of avian influenza subtypes using fluorescent imaging[49]. From the previous study, it has become apparent that a versatile host response needs to be measured to better understand the infection process at the individual cell level.

We were intrigued by the possibility of using the ER stress-responsive fluorescence of EliF molecules to distinguish the early infection stage of avian influenza (AI) in host cells. The current gold standard protocol for AI virus diagnosis involves a two-step

workflow[50]. In this method, the egg-based virus amplification step is required to obtain hundreds of microliters of sample volume for hemagglutination assay (HA). Using ROS chemosensors, we have previously demonstrated that the cell-based assay has significant advantages over conventional assay: cell culture is more cost-effective than egg-based amplification; single-cell-based assay is more sensitive than serologic assay[49]. Since molecular probes capable of distinguishing infected cells by distinct cellular mechanisms have enormous value, we tested the practical utility of EliF dyes for an early diagnosis of AI infection in a cell-based assay.

To validate cell-based AI infection profiling, three cell lines (HeLa, SK-OV3, and SH-SY5Y) exhibiting significantly different infectability patterns were chosen (Fig. 8a). Among the final two candidates, **EliF-2c** was employed as the probe because of its stronger fluorescence response than **EliF-1b** under ER stress conditions (Fig. 7); an added bonus was its lower photo-cytotoxicity (Supplementary Fig. 19). During the infection process, changes in the emission were monitored every 12 h at the single-cell level on flow cytometry (Fig. 8b, c). After 24 h, a significant fluorescence enhancement was observed only in HeLa cells with a high infectability level. Without any ER stress-inducing agents, influenza infection triggered host cell response that altered fluorescence emission of **EliF-2c**.

Encouraged by this distinctive response depending on the infection level, we further examined additional model cells from respiratory-related organ origin: human tracheal epithelial cell (HTEpC), and nasal septum origin carcinoma cell (RPMI 2650). Normal cells from humans (HUVEC) were also subjected to the test. We found that **EliF-2c** consistently presented clear fluorescence increment in these cells (Supplementary Fig. 30). It is also noteworthy that the cellular expression of major target proteins was also upregulated upon AI infection (Supplementary Fig. 31). Considering that the conventional egg-based assay requires 3–5 days for virus amplification, **EliF-2c** can shorten the diagnosis time to 24 h by detecting early AI infection at the single-cell level. These results clearly demonstrate that ER stress responding EliF can be used to monitor the early-stage of AI infection.

Our observation of ER stress induction in early AI virus infection is consistent with the previous report on differential transcriptome changes of several RNA virus infections[51]. Interestingly, we found that three proteins (NCAPG, PSPH, and PDXDC2P) exhibiting the highest $T_m$ shifts for EliF also showed significant expression perturbation upon RNA virus infection. In addition, a recent multi-omics analysis of unfolded protein response also revealed that ER stress induces the up-regulation of NCAPG, PDXDC1, and PSPH, which is consistent with our observation[52]. Taken together, we postulate that **EliF-2c** responds to ER stress by interacting with these proteins. A detailed molecular-level understanding of this tantalizing prospect will be a topic of future studies.

We have constructed π-extended benzimidazole fluorophores that localize selectively at ER, respond to ER stress, and detect AI virus infection. Our synthesis involves modular construction of the core π-skeleton, and stepwise installation of {BF₂}⁺ units to diversify the structure. A balancing tradeoff between environment-sensitivity and fluorescence quantum yield was observed, when the N−H···N hydrogen bonds were replaced consecutively by the {N−BF₂−N} dative bonds.

Even without specific subcellular organelle-targeting functional groups, certain EliF molecules are taken up by living cells and localized selectively at ER. Moreover, they also detect ER stress by changing the fluorescence characteristics of individual cells. To the best of our knowledge, EliF represents the only class of ER-targeting and ER stress-responding molecular probes that diagnose the early-stage of AI virus infection within a day by avoiding lengthy and complicated sample preparation steps.

Influenza infection is a highly heterogeneous process even in homogeneously cultured cells. As such, fractionation of infected cells at the single-cell level, despite its technical challenge, has a significant

impact on elucidating the mechanism of influenza infection, possibly by employing multi-omics techniques. We have demonstrated the robustness of **EliF-2c** in FACS-based counting assay or fractionation sorting assay, which is an important first step to obtain detailed information on influenza infection. Efforts are currently underway in our laboratories to establish robust cell-based AI virus screening applications with structurally more elaborate fluorophores.

# Methods

Synthetic procedures and characterization of EliF fluorophores reported in this work are provided in Supplementary Information.

## Physical measurements

$^{1}$H NMR, $^{13}$C NMR, and $^{19}$F NMR spectra were recorded on a 300 MHz Bruker Advance DPX-300, a 400 MHz Agilent 400-MR DD2 Magnetic Resonance System, or a 500 MHz Varian/Oxford As-500 spectrometer. Chemical shifts were referenced to the residual solvent peaks[53]. High-resolution electrospray ionization (ESI) mass spectra were obtained on an ESI-Q-TOF mass spectrometer (Compact, Bruker Daltonics Inc). FT-IR spectra were recorded on a Shimadzu IRTracer-100 FT-IR Spectrophotometer. Fluorescence spectra were recorded on a Photon Technology International Quanta-Master 400 spectrofluorometer with FelixGX software. Quantum yields were determined by using an integrating sphere attached to the instrument. Time-resolved photoluminescence (PL) decay measurements were carried out with an Edinburgh FS5-TCSPC equipped with a 375 nm diode laser (EPL-375). Elemental analysis was performed by a PerkinElmer 2400 Series II CHNS/O Analyzer. Single-crystal X-ray diffraction studies were carried out using Cu Kα radiation on an XtaLAB AFC12 (RINC): Kappa dual home/near diffractometer or a SuperNova, Dual, Cu at zero, AtlasS2 diffractometer.

Bright-field and fluorescence images were taken in Leica DMi8 microscope with Leica Application Suite X (LAS X Version: 3.7.0.20979) software or in Nikon Eclipse Ti microscope (Nikon, G9241) with NIS-Elements BR (version 4.50) software. Cytotoxicity and photo-cytotoxicity were analyzed with SpectraMax microplate reader (Molecular Devices, Sunnyvale, CA, USA) and SoftMax Pro software (version 5.4). Data analysis including statistical analysis was performed using OriginPro 2019 (version 9.6.0.172) or R (version 4.1.1).

## Computational studies

Ground-state and excited-state geometry optimization of EliF molecules were carried out with density functional theory (DFT) and time-dependent density functional theory (TD-DFT) calculations using B3LYP density functional with a 6–31 G(d) basis set. Solvation was considered for the excited-state geometry optimization using self-consistent reaction field (SCRF) calculations with conductor-like polarizable continuum model (CPCM) of $CHCl_3$ ($\varepsilon = 4.71$)[54]. Vibrational frequency calculations were performed to confirm that all optimized geometries converged to minima with no imaginary frequencies or saddle points for the transition state. The entire calculations were performed with Gaussian 16 program suite[55]. Cartesian coordinates of the geometry optimized structures are provided as a Source Data file.

## General information on used cell lines, antibodies, and primers

HeLa (Korean Cell Line Bank, KCLB No. 10002), SK-OV3 (Korean Cell Line Bank, KCLB No. 30077), SH-SY5Y (Korean Cell Line Bank, KCLB No. 22266), Human Tracheal Epithelial Cells (HTEpC) (PromoCell, cat. no. C-12644), Human Umbilical Vein Endothelial Cells (HUVEC) (Cell Engineering For Origin, cat. no. CEFOgro-HUVEC), and RPMI 2650 (Korean Cell Line Bank, KCLB No. 10030) cells were obtained from the indicated suppliers and used throughout the study.

Anti-matrix protein 1 (M1) antibody (mouse; Abcam, cat. no. ab22396; clone number: GA2B) and anti-mouse Alexa488 (rabbit;

Thermo Fisher Scientific, cat no. A-11011; RRID: AB_143157) antibodies were obtained from commercial manufacturers and used for immunocytochemistry analysis.

Primers used in a polymerase chain reaction (PCR) were NCAPG (F-TTCAAGGCTGGTTACGGTTC, R-AGGGCACACCAATACAAAGC), PDXDC1 (F-TAGGAACTCGGGGAGAGGAT, R-CCTTCGGGATC TGATT TCAA), PDXDC2P (F-AGATGGATGTTGCCTTCCTG, R-CCAGCACTGAT GAGGAGACA), PSPH (F-GACAGCACGGTCATCAGAGA, R-CCTGAACA TTTCGCTCCTGT), and ACTB (F-TGACGGGG TCACCCACACTG, R-CT AGAAGCATTTGCGGTGGA), and purchased from Macrogen, Inc.

## Mammalian cell culture and fluorescent live-cell imaging

HeLa cells were grown in DMEM supplemented with 10% FBS, 100 units/mL penicillin, and 100 µg/mL streptomycin in a humidified atmosphere containing 5% $CO_2$ at 37 °C. To screen live-cell imaging applicability, each EliF probe (10 µM) was incubated in the HeLa cells for 30 min (final DMSO concentration = 1%). Prior to obtaining fluorescence images, cells were washed twice using fresh DMEM media. Images were obtained under different microscope filter settings (DAPI, $\lambda_{exc} = 325–375$ nm, $\lambda_{em} = 435–485$ nm; F1, $\lambda_{exc} = 325–375$ nm, $\lambda_{em} = 512–542$ nm; F2, $\lambda_{exc} = 325–375$ nm, $\lambda_{em} = 565–605$ nm; FITC, $\lambda_{exc} = 460–500$ nm, $\lambda_{em} = 512–542$ nm; RHOD, $\lambda_{exc} = 541–551$ nm, $\lambda_{em} = 565–605$ nm) using Leica DMi8 microscope (Leica Inc., Germany) with ×40 dry lens.

For the investigation of the subcellular organelle localization, HeLa cells were plated in a black transparent 96-well plate. To stain cellular organelles, cells were transfected with CellLight™ organelle trackers for 24 h. Different organelle trackers were used, namely, CellLight™ ER-RFP (Thermo Fisher, C10591), CellLight™ Mitochondria-RFP (Thermo Fisher, C10505), CellLight™ Golgi-RFP (Thermo Fisher, C10593), CellLight™ Lysosomes-RFP (Thermo Fisher, C10597), and CellLight™ Peroxisome-GFP (Thermo Fisher, C10604). Following the transfection, cells were treated with probes (10 µM) for organelle localization imaging (final DMSO concentration = 1%). The nucleus was stained with NUCLEAR-IDr Red DNA stain (Enzo, ENZ-52406; $\lambda_{exc} = 590–650$ nm, $\lambda_{em} = 662–738$ nm) for 30 min. After incubation, cells were washed twice with fresh DMEM media, and the fluorescence images of the probes and organelle trackers were captured by using a Leica DMi8 microscope (Leica Inc., Germany) with a ×40 dry lens. The image acquisition settings were optimized by providing the excitation light from a mercury lamp (ECL6000, Leica Inc., Germany) with mild intensity (25%) during imaging. The images were processed in Leica Application Suite X (LAS X Version: 3.7.0.20979) software. The co-localization analysis of the indicated lines in the images was performed by using Fiji/ImageJ (version 1.46r).

## Cell culture, heating, and cell lysis for thermal proteome profiling

HeLa cells were grown in DMEM supplemented with 10% FBS, 100 units/mL penicillin, and 100 µg/mL streptomycin in a humidified atmosphere containing 5% $CO_2$ at 37 °C. HeLa cells were plated in a 10 cm cell culture dish for 24 h, and treated with 10 µM of each EliF probe (**EliF-1b** and **EliF-2c**) for 30 min (final DMSO concentration = 1%). To harvest the cells, trypsin (0.05%) was added to the plate. After incubation at 37 °C for 2 min, the reaction was terminated by PBS. The resulting samples were pelleted and resuspended in PBS to a final count of $2 \times 10^6$ cells/mL. The samples were pelleted again and resuspended in 1.1 mL of cold PBS supplemented with protease inhibitors. The cell suspensions were transferred (100 µL each to ten PCR tubes), and centrifuged at 300 g for 3 min at $T = 4$ °C to pellet the cells. After removing 80 µL of PBS, each 20 µL of the concentrated cells were heated at ten designated temperatures (37, 41, 44, 47, 50, 53, 56, 59, 63, and 67 °C) for 3 min in C1000 Touch Thermal Cycler (Bio-Rad, 1841000). After heating, the samples were incubated in the PCR tubes at r.t. for 3 min. The cells were lysed by freeze-thaw cycles and

centrifuged at 20,000×$g$ for 20 min at $T$ = 4 °C. Finally, 80 μL of each supernatant with the soluble protein fraction was carefully transferred to a new tube for further analysis.

## In-solution protein digestion

Cell lysate samples were mixed with 100 μL of 8 M urea in 3 K filter units (Millipore, UFC500396), and centrifuged at 14,000×$g$ for 15 min. The step was repeated once. After collecting the supernatants, dithiothreitol (10 mM) was added to the filter units and incubated for 40 min at 37 °C. Subsequently, iodoacetamide (50 mM) was added to the filter units and incubated for 20 min in dark. Final buffer exchange was performed by adding 100 μL of ammonium bicarbonate buffer (50 mM), followed by a 14,000×$g$ spin for 15 min. After two additional buffer exchanges, the samples were digested with trypsin (Promega, V5280) overnight at 37 °C. The trypsin digestion was terminated with 1% formic acid, and the samples were desalted with C18 cartridges (Waters, WAT054955). The C18 cartridges were conditioned with acetonitrile, and equilibrated with 0.1% formic acid. The peptides bound to the C18 cartridges were washed with 0.1% formic acid and eluted by elution buffer (70% acetonitrile and 0.1% formic acid).

## TMT-10-Plex labeling and off-line fractionation at basic pH

Digested peptides were resuspended in 100 μL of triethylammonium bicarbonate buffer (100 mM). Each sample was mixed with a separate TMT (Thermo Scientific, 90110) label suspended in 41 μL of acetonitrile. Samples were incubated at r.t. for 1 h. Labeling was quenched with 8 μL of 5% hydroxylamine for 15 min. Samples were combined and dried in SpeedVac. The labeled peptides were resuspended in 300 μL of 0.1% formic acid and desalted by C18 cartridges with standard procedures. The C18 cartridges were conditioned with acetonitrile and equilibrated with 0.1% formic acid. The peptides bound to the C18 cartridges were washed with 0.1% formic acid and eluted by elution buffer (70% acetonitrile, 0.1% formic acid). The samples were separated by XBridge BEH130 C18 column (Waters, 186003128); the basic pH fraction of the eluent was gradually increased from 3 to 60% over 85 min; flow rate = 40 μL/min. The eluted fractions were collected for each 1 min into low binding tubes. After collecting the first 20 fractions, the 21st fraction was collected into the same tube as fraction #1, the 22nd fraction was collected into fraction #2, and so on. As a result, four fractions, i.e., $i$th, $(i + 20)$th, $(i + 40)$th, and $(i + 60)$th, were collected into each tube.

## LC-MS/MS analysis

All LC-MS/MS data were collected by an UltiMate™ 3000 RSLCnano System (Thermo Scientific, 5200.0356) equipped with a 50-cm C18 analytical column (Thermo Scientific, ES903), a 2-cm trap column (Thermo Scientific, 164535), an Orbitrap Eclipse Tribrid mass spectrometer (Thermo Scientific, FSN04−10000), and Xcalibur software (version 4.1). The analytical column temperature was maintained at 50 °C. Each fraction was eluted by using a 180 min method over a gradient from 5 to 80% acetonitrile in 0.1% formic acid. Full-scan MS spectra (400−2,000 m/z) were acquired in the FT-Orbitrap at a resolution of 120,000, followed by the selection of the twenty most intense ions for HCD-MS2 fragmentation. Directly after each MS2 experiment, the ten most intense fragment ions of the precursor $m/z$ were selected for HCD-MS3. The resulting MS/MS/MS data were searched using Uniprot (Taxonomy: Homo sapiens, 2020.05.19) and analyzed by Proteome Discoverer software (v.2.4). TMT-plex (K) and Carbamidomethylation (C) were set as static modifications, and oxidation (M) and acetyl (N-terminus) were set as variable modifications. The precursor and fragment mass tolerances were set to 10 ppm and 0.6 Da, respectively. The maximum peptide and site false discovery rates were specified as 0.01 using Percolator.

## Fluorescence responses of EliF-2c in the monitoring of cells infected with avian influenza

To monitor AI infection at the early-stage, different cells were infected with the H1N1 virus (EID$_{50}$/mL; $10^{7.167}$) for 12 and 24 h, and treated subsequently with **EliF-2c** (10 μM) for 30 min. The cells were trypsinized, harvested, and centrifuged for 5 min at 136×$g$. The supernatants were discarded, and the remaining cell pellets were resuspended in PBS. The resulting samples were subjected to flow cytometry analysis on a Gallios Flow Cytometer (Beckman Coulter, USA) by counting 10,000 events for each sample using Kaluza software (version 2.1). The fluorescence signals were recorded by FL10 ($\lambda_{exc}$ = 405 nm and $\lambda_{em}$ = 530−570 nm). The data was processed with FlowJo Single-Cell Analysis Software v10.7.2 (FlowJo, LLC, Ashland, OR, USA) by using histogram offset plots. The fluorescence intensity fold changes were calculated by comparing the virus-treated and control groups.

## Cell line susceptibility profiling of H1N1 virus using immunocytochemistry

For avian influenza (AI) infection profiling, HeLa, SK-OV3, and SH-SY5Y cells were used. Cells were seeded in a 96-well plate ($1 \times 10^4$ cells/well) for 24 h and infected with AI virus strain H1N1 (A/NSW/33) in a 1/10-fold serial dilution starting with Embryo Infectious Dose 50 value (EID$_{50}$/mL) $10^{7.167}$ for 24 h[49]. For the staining of AI-infected cells, a 1:100 dilution of anti-matrix protein 1 (M1) antibody (mouse; Abcam, cat. no. ab22396; clone number: GA2B) was used. Prior to antibody staining, the infected cells were fixed with 100% methanol for 10 min, and permeabilized by using 0.25% Triton-X100 for 10 min. After blocking with 2.5% BSA for 1 h with gentle shaking, cells were immunostained with an anti-M1 antibody for 2 h. Unbound antibodies were washed out with PBS-T (PBS containing 0.1% Tween-20) three times and incubated 1:3000 dilution of anti-mouse Alexa488 (rabbit; Thermo Fisher Scientific, cat no. A-11011; RRID: AB_143157) for 1 h. The cells were subsequently washed with PBS-T three times, and the nuclei were stained by Hoechst34580 for 15 min. After three additional washings with PBS-T, immunocytochemistry (ICC) was performed with Operetta High Contents Image System (PerkinElmer) and analyzed by Harmony 3.1 software (PerkinElmer).

## Reporting summary

Further information on research design is available in the Nature Research Reporting Summary linked to this article.

# Data availability

The X-ray crystallographic coordinates for structures reported in this study have been deposited at the Cambridge Crystallographic Data Centre (CCDC), under deposition numbers 2003264 (**EliF-1a**), 2003269 (**EliF-1c**), 2003265 (**EliF-1d**), 2003263 (**EliF-2a**), 2003267 (**EliF-2b**), 2003268 (**EliF-2d**), and 2003266 (**EliF-3d**). These data can be obtained free of charge from The Cambridge Crystallographic Data Centre via www.ccdc.cam.ac.uk/data_request/cif. The MS/MS/MS data obtained in this study were searched using Uniprot (Taxonomy: Homo sapiens) [https://www.uniprot.org/uniprotkb?facets=reviewed:true&query=(taxonomy_id: 9606)]. The proteomics dataset generated in this study is provided in Supplementary Data 1 and available in the MassIVE repository under accession code MSV000089586. All data that support the findings of this study are available from the corresponding authors upon request. Supplementary methods, NMR spectra, supplementary table, and supplementary figures are available in the Supplementary Information. Supplementary Data 1 includes the results of thermal proteome profiling experiments, a list of the primary hit proteins and their $T_m$ shifts, and results of gene ontology enrichment analysis. Source data are provided with this paper.

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

## Acknowledgements

This work was supported by the Basic Research Grant (2020R1A2C2006381 to D.L.) and the Bio & Medical Technology Development Program (2018M3A9H4079286 to J.-S.L.) through the National Research Foundation of Korea (NRF). D.L. also acknowledges the financial support from the Samsung Science and Technology Foundation (SSTF-BA1701-09). Dr. Haque is a recipient of the Korea Research Fellowship Program (NRF-2018H1D3A1A02074556). T.K. thanks the NRF-2016-Global Ph.D. Fellowship Program (2016H1A2A1906550).

## Author contributions

T.K., M.M.H., and B.L. contributed equally to this work. T.K., B.L., and D.L. initialized the project. T.K., B.L., Y.K., and J.L. synthesized and characterized the compounds. D.L. supervised the experimental component. M.M.H. and S.C.H. performed fluorescence imaging and FACS experiment, and K.H. conducted proteomics analysis under the guidance of J.-S.L. All authors contributed to drafting the manuscript.

## Competing interests

The authors declare no competing interests.
