## [Peer Review File · Nature Communications]

Orthogonally-Tunable and ER-Targeting Fluorophores Detect Avian Influenza Virus Early InfectionREVIEWER COMMENTS

Reviewer #1 (Remarks to the Author):

The paper is interesting, reporting new T-shaped benzimidazole compounds for orthogonally-tunable and ER-targeting fluorophores detect avian influenza 2 virus early infection in single cell level. However, the current manuscript represent a sufficient basis for publication in Nature Communication at the present time and that further experimental work and clarifications are required:

1. The current fluorescence spectra shown in Fig. 4 and Supplementary Fig. 8, it is impossible to compare the difference in emission intensity since the author normalized the emission intensity.
2. For the in vitro assay, normal respiratory tract cells should be used as a human cell model for influenza virus infection and proteomics research rather than using HeLa cancer cells.
3. ER stress can be induced by chemical compound as well, the author should also clarify that the fluorescence enhancement was indeed due to avian influenza 2 virus infection rather than ER stress induced by other drugs in normal human cells.
4. The author used 350 nm UV light to excite the complex, which may cause damage to the cells and change in ER stress. Thus it will be better to confirm the photo-cytotoxicity of the dyes. Meanwhile, the emission wavelength of the compound (527 nm) was also too short for in vivo infection detection.
5. The in vivo model assay should be added to confirmed the avian influenza 2 virus detection in cells.

Reviewer #2 (Remarks to the Author):

Pyridyl substituted benzimidazoles have been synthesized and converted to boron complexes. The compounds are interesting however are not surprising. Though the synthesis of the compounds are unambiguous, the fluorescence data have not been explained appropriately. The dual emission observed for EliF-1a in most of the solvents (Figure 4a) is not properly addressed. The emissive states corresponding to this dual emission must be ascertained. Such a dual emission is absent in EliF-1c however, is present in the most polar solvent for EliF-1b. This anomalous behavior also require an explanation. Solvatochromism data for other compounds containing different substituents must be presented in the supplementary information and explained appropriately. The excited state lifetime of the compounds must be presented in the manuscript. HOMO and LUMO of EliF-1b is missing in the supplementary information (Figure S10). Similarly HOMO and LUMO of other compounds (1a, 1c, 1d, 2a, 2c, 2d, etc.) is missing. It is also interesting to see how the HOMO and LUMO is there when diethylamine group is present in the molecule. So there is no better understanding on the emission behavior of the molecules presented in the manuscript prior to describing their application. There are several important information missing about the emission characteristics of the compounds presented in the manuscript. To justify the use of CN derivative in the application the emission properties of other compounds should be decently detailed in the manuscript.

Reviewer #3 (Remarks to the Author):

This report is focused mainly on the disclosed crystal structures. Upon visual inspection, analysis of the cif, and checkCIF reports, the deposited structures appear to be satisfactory. The main comment that I have is that the moiety and sum formulas between the reported and calculated values in the checkCIF report. One structure (elif-et_1) has an oxygen in the formula where there isn't one. In many of the structure, the Z' needs to be updated prior to updating the moiety and sum formulas. The SC-XRD instruments used should also be listed in the "Physical Measurements" section.

For the computed structures, are they a global (or local) minimum or first-order saddle points (transition states) or higher (meaningless)? Were frequency calculations performed to verify this? If not, please do so, and report the Cartesian coordinates of the computed structure in the supporting information.

Reviewer #1. We thank the reviewer for careful reading and detailed evaluation of our manuscript. Listed below are our point-by-point responses to her/his critical comments and helpful suggestions.

1. The current fluorescence spectra shown in Fig. 4 and Supplementary Fig. 8, it is impossible to compare the difference in emission intensity since the author normalized the emission intensity.

- According to the reviewer's suggestion, we added unprocessed fluorescence spectra (i.e., without normalization) of the EliF series molecules as **Supplementary Fig. 10** in the revised SI. This data provides detailed information on the environment-sensitive nature of the fluorescence signal as changes in the emission wavelength and intensity in different solvents.

Supplementary Fig. 10. Solvent polarity-dependent changes in the relative fluorescence intensities of (a) **EliF-1b**, (b) **EliF-2b**, (c) **EliF-3b**, (d) **EliF-1d**, (e) **EliF-2d**, (f) **EliF-3d**, (g) **EliF-2a**, (h) **EliF-2c**, and (i) **EliF-3a** in cyclohexane, Et₂O, CHCl₃, DMSO, MeCN, iPrOH, EtOH, and H₂O (T = 293 K; A_{exc} = 350 nm).

. For the sake of completeness, we also report solvent polarity-dependent changes in the normalized emission spectra of **EliF-2a**, **EliF-2c**, and **EliF-3a** as **Supplementary Fig. 9**. Please see our response to **Reviewer #2's Point #3** below for an in-depth theoretical explanation of the solvatochromism.

2. For the in vitro assay, normal respiratory tract cells should be used as a human cell model for influenza virus infection and proteomics research rather than using HeLa cancer cells.

- We appreciate the reviewer's comments on the choice of cell model, which prompted our further investigation to significantly improve the manuscript's quality and completeness (see below). Before going into that part, we would like to explain briefly why we chose the HeLa cell for our initial exploratory studies. We previously examined 23 cell lines for infection profiling against three avian influenza subtypes (Angew. Chem. Int. Ed. 2018, 57, 9716–9721). In the follow-up studies, the infection profiling has been expanded further to include 35 additional cell lines (unpublished results). Based on the results from these exploratory studies, we chose three cell lines (SK-OV3, SH-SY5Y, and HeLa) that displayed distinctive infection patterns against the H1N1 virus used in the current investigation (**Fig. 8a**). With the influenza virus, HeLa cells showed high infection, to which **EliF-2c** exhibited a unique fluorescence turn-on response. Moreover, our initial screening with the EliF probes

for live-cell imaging was carried out with HeLa cells. Accordingly, proteomic studies to identify the target were also conducted with HeLa cells to fully establish the relationship between ER-specific fluorescence imaging and interactome protein identifications.

- We fully agree with the reviewer that normal respiratory-related cells are more relevant models for the diagnostic tests of influenza virus infection. For this revision, we thus carried out extensive studies with three additional cell lines: (1) human tracheal epithelial cells (HTEpC) that the reviewer specifically suggested; (2) RPMI 2650 as nasal septum originated respiratory-related cancer cell; (3) HUVEC as normal human cell. As summarized in **Supplementary Fig. 30** of the revised SI, FACS-based response data unambiguously demonstrates consistent fluorescence increment with **Elif-2c** across all these cell lines.

Supplementary Fig. 30. Flow cytometry analysis of **Elif-2c** ($10 \mu\text{M}$) in H1N1 virus infected HTEpC, RPMI 2650, and HUVEC cells using FL10 filter ($A_{\text{exc}} = 405 \text{ nm}$; $A_{\text{em}} = 550/40 \text{ nm}$).

- . To report these new findings, the following sentence was added to the revised manuscript:

"Encouraged by this distinctive response depending on the infection level, we further examined additional model cells from respiratory related organ origin: human tracheal epithelial cell (HTEpC), and nasal septum origin carcinoma cell (RPMI 2650). Normal cells from human (HUVEC) were also subjected to the test. We found that **Elif-2c** consistently presented clear fluorescence increment in these cells (Supplementary Fig. 30)."
(Page 15, Paragraph 1)

We are truly grateful to the reviewer for her/his well-intended criticism and suggestion to use normal and respiratory tract cell models. Our studies further demonstrate the robustness of **Elif-2c** for the assay, and significantly improved the quality of the manuscript.

3. ER stress can be induced by chemical compound as well, the author should also clarify that the fluorescence enhancement was indeed due to avian influenza 2 virus infection rather than ER stress induced by other drugs in normal human cells.

- The reviewer raised concerns regarding the potential *crossstalk effect* of ER stress and virus infection. In our studies, two separate experiments were carried out: response to ER stress (**Fig. 7**); response to virus infection (**Fig. 8**). In **Fig. 7**, fluorescence response was measured upon treating ER stress-inducing chemicals (thapsigargin or tunicamycin) without virus infection. In **Fig. 8**, fluorescence response was measured upon influenza virus infection

in the absence of ER stress inducers. Therefore, the fluorescence enhancement of **EliF-2c** presented in **Fig. 8** arises solely from a viral infection, without interference from ER stress-inducing drugs. To avoid unintended confusion, we further clarified this point in the main text, which reads:

"Without any ER stress-inducing agents, influenza infection triggered host cell response that altered fluorescence emission of **EliF-2c**."
(Page 14, Paragraph 3)

4. The author used 350 nm UV light to excite the complex, which may cause damage to the cells and change in ER stress. Thus it will be better to confirm the photo-cytotoxicity of the dyes. Meanwhile, the emission wavelength of the compound (527 nm) was also too short for in vivo infection detection.

. The potential photo-toxicity of fluorescent probes is a legitimate concern. In the initially submitted version of the manuscript, we only reported cytotoxicity assay without UV-light exposure. According to the reviewer's suggestion, photo-cytotoxicity was evaluated and reported in **Supplementary Fig. 19**. Overall, **EliF-1** series become more cytotoxic upon UV-light ($\lambda = 350$ nm) exposure. **EliF-2a** and **EliF-2d** showed notable toxicity only at the highest dose (30 μ M). In general, the photo-toxicity diminishes with increasing level of borylation: **EliF-1** >> **EliF-2** > **EliF-3**). This intriguing empirical observation is briefly discussed in the main text as:

"While the **EliF-1** series molecules become more toxic under UV irradiation ($\lambda = 350$ nm) conditions, photo-toxicity diminishes with increasing level of borylation: **EliF-1** >> **EliF-2** > **EliF-3** (Supplementary Fig. 19)."

(Page 8, Paragraph 2)

Supplementary Fig. 19. Cellular toxicity effect of EliF probes measured by CellTiter-Glo assay. Cellular viability of HeLa cells was determined by incubation with each EliF probe for 30 min followed by 350 nm UV irradiation for 10 s according to the manufacturer's protocol. Each compound was examined as triplicates, and the mean and standard deviation are plotted (** for $p < 0.01$; two-tailed paired t-test). Shown below are representative bright field images of the cells.

5. The *in vivo* model assay should be added to confirmed the avian influenza 2 virus detection in cells.

- The reviewer questioned the practical utility of *in vitro* FACS-based assay. For imaging probes that aim to visualize virus infection in living organisms, *in vivo* model is a crucial assessment factor. However, influenza infection is a highly heterogeneous process, even with a homogeneously cultured cell line. As such, fractionation of infected cells at the single-cell level, despite its technical challenge, has a significant impact on elucidating the mechanism of influenza infection, possibly by employing multi-omics techniques. For such a purpose, we believe that FACS is the optimum platform to evaluate infections at the single-cell level. **EliF-2c** based counting assay or fractionation sorting assay is an important first step that should provide detailed information on influenza infection. To make this point clear, the following sentences were added to the main text:

?!Influenza infection is a highly heterogeneous process even in homogeneously cultured cells. As such, fractionation of infected cells at the single-cell level, despite its technical challenge, has a significant impact on elucidating the mechanism of influenza infection, possibly by employing multi-omics techniques. We have demonstrated the robustness of **EliF-2c** in FACS-based counting assay or fractionation sorting assay, which is an important first step to obtain detailed information on influenza infection.?!

(Page 16, Paragraph 4)

Reviewer #2. We thank the reviewer for thoughtful comments and helpful suggestions on the mechanism and kinetics of de-excitation, interpretation of photophysical data, electronic structures of the frontier molecular orbitals (FMOs), and selection of the probe molecule for eventual biological applications. Listed below are our point-by-point responses.

1. The dual emission observed for EliF-1a in most of the solvents (Figure 4a) is not properly addressed. The emissive states corresponding to this dual emission must be ascertained.

- Unlike the reviewer's comment, **EliF-1 a** does NOT show dual emission. The figure panel **Fig. 4a** shows the dual emission behavior of **EliF-1b**, which we will explain in detail. Comprised of benzimidazole and pyridine units to function as intramolecular hydrogen bond donor-acceptor (HBD-HBA) pair, **EliF-1b** can potentially take an excited-state intramolecular proton transfer (ESIPT) pathway to produce dual emission. This mechanistic postulation involves tautomer equilibrium between the normal form (N*) and tautomeric form (T*) (see **Fig. R1** below). To check the validity of this excited-state dynamics, we proceeded to carry out time-dependent density functional theory (TD-DFT) computational studies.

Fig. R1. Tautomer equilibrium of **EliF-1b** by excited-state intramolecular proton transfer (ESIPT) reaction.

- As shown in Supplementary Fig. 12 in the revised SI, the geometries of N* and T* tautomers were optimized in the S₁ state. Since T* is lower in energy than N* ($\Delta E_{N^* \rightarrow T^*} = -0.1$ eV), ES IPT is a thermodynamically favored process in the excited state. In addition, calculated oscillator strengths of N* ($f = 0.5538$) and T* ($f = 0.1727$) indicate that both tautomers are emissive. The large activation energy barrier (0.48 eV) of the N*~T* tautomerization, computed by optimizing the transition state geometry, suggests that ES IPT is kinetically controlled. Consequently, N* has a sufficient lifetime to emit light (as local emission, LE) before undergoing ES IPT to T*.

Supplementary Fig. 12. Energy diagram of EliF-1b in the S₁ excited-state and S₀ ground state calculated by TD-DFT (CPCM/B3LYP/6-31G(d) level of theory).

- To obtain experimental evidence for the excited-state dynamics involving N* and T*, we carried out fluorescence lifetime measurements by time-correlated single-photon counting (TCSPC) technique. Comprised of relatively fast ES IPT and relatively slow de-excitation components, the fluorescence decays of N* and T* emissions typically take double exponential forms reflecting depopulation/population of N* and T*.¹ However, when the energy difference and the activation barrier between N* and T* are large, the ES IPT becomes under kinetic control. In such a situation, the depopulation rate of N* by the forward ES IPT reaction becomes sufficiently slow to compete with the de-excitation of N* to the ground state. As a consequence, the decay of N* emission becomes single exponential with a short lifetime.
- As summarized in Supplementary Table 1 of the revised SI, EliF-1b shows a characteristic fluorescence decay profile of kinetically controlled ES IPT, with a relatively short fluorescence lifetime ($\tau = 0.50$ ns) of N* emission ($\lambda_{em} = 440$ nm) and a double exponential fitted decay ($\tau = 0.22$ and 1.47 ns) of T* emission ($\lambda_{em} = 530$ nm). The negative pre-exponential factor (-0.110) of the shorter lifetime component ($\tau = 0.22$ ns) of T* emission indicates that it is resolved as the rise time of T* species. A similar fluorescence decay profile was observed for EliF-1d displaying dual emissive behavior.
- We thus conclude that the ES IPT process is responsible for the dual emission of EliF-1b, which the reviewer inquired into. By reshaping the energy coordinates of ES IPT reaction, solvent polarity (as well as HBD/HBA ability) directly impacts the intensity ratio of the

shorter-wavelength (from N* state) and longer-wavelength (from T* state) components of dual emissive EliFs. To explain the molecular mechanism of dual emission, the following sentences were added to the main text:

"The most noticeable change is a complete disappearance of the dual emission (Fig. 4a and Supplementary Fig. 8a) from ESIPT. For **EliF-1** having N–H...N hydrogen bond, the ESIPT reaction is under kinetic control (Supplementary Fig. 12), allowing both tautomers to take radiative pathways to produce dual emission. This theoretical prediction is supported by a relatively short fluorescence lifetime of the local emission and a double-exponential fitted decay of the tautomer emission of **EliF-1b** (Supplementary Table 1)."

(Page 5, Paragraph 4)

We thank the reviewer for prompting in-depth computational/experimental studies to elucidate the underlying molecular mechanism of the dual emission, which significantly improved the quality of the manuscript.

2. Such a dual emission is absent in EliF-1c however, is present in the most polar solvent for EliF-1b. This anomalous behavior also require an explanation.

- . Looking at the data provided in **Fig. 4**, we suspect that the reviewer actually meant **EliF-3b** instead of **EliF-1c** (NOTE: **EliF-1c** is NOT reported in **Fig. 4**); **EliF-2b** instead of **EliF-1b** (NOTE: the dual emission of **EliF-1b** originates from ESIPT, which is NOT anomalous at all; please see our response to **Point #1** above).
- . First, we thank the reviewer for a careful reading of our manuscript. Borylated EliF lacks HBD–HBA pairs for ESIPT (see our response to **Point #1** above). Therefore, the apparent dual emission of borylated **EliF-2b** only in water (but not in other solvents), shown in previous **Fig. 4b**, is indeed quite anomalous and counterintuitive.
- . While repeating the experiments and checking the reproducibility, we realized that the longer wavelength emission ($A_{\max,em} = 530 \text{ nm}$) of **EliF-2b** becomes more pronounced at higher sample concentrations. Concentration-dependent measurements (see **Fig. R2** on the next page) of $[\text{EliF-2b}] = 0.5\text{--}10 \text{ }\mu\text{M}$, in conjunction with dynamic light scattering (DLS) studies of the most concentrated sample (10 μM ; see **Fig. R3** on the next page), confirmed the formation of solution aggregates in water to produce longer-wavelength emission. Without a polar N–H bond or intrinsic positive charge, the **EliF-2b** suffers from low solubility in water and readily aggregates upon concentration.
- . To avoid artifacts from solution aggregates, emission spectra of all EliF molecules in water were re-recorded at dilute sample concentration of 1.0 μM . Revised **Fig. 4** in the main text and **Supplementary Fig. 8** in the SI reflect these changes.

Fig. R2. Normalized fluorescence spectra of **EliF-2b** in water at different concentrations (0.5–10 μM); $T = 293 \text{ K}$.

Fig. R3. Dynamic light scattering (DLS) profile of **EliF-2b** (10 μM) in water at $T = 293 \text{ K}$.

Fig. 4 | Solvent-dependent changes in the fluorescence spectra. Normalized fluorescence spectra of **EliF-1b** (a), **EliF-2b** (b), and **EliF-3b** (c) in cyclohexane, Et₂O, CHCl₃, DMSO, MeCN, iPrOH, EtOH, and water (from front to back; corresponding to increasing solvent polarity as denoted by the $\epsilon_T(30)$ values¹⁸ in the y-axis; $T = 293 \text{ K}$; $\lambda_{\text{exc}} = 350 \text{ nm}$).

Supplementary Fig. 8. Solvent polarity-dependent changes in the normalized fluorescence spectra of (a) **EliF-1d**, (b) **EliF-2d**, and (c) **EliF-3d** in cyclohexane, Et₂O, CHCl₃, DMSO, MeCN, iPrOH, EtOH, and water (from front to back; corresponding to increasing solvent polarity as denoted by the E_T(30) values in the y-axis; T = 293 K; A_{exc} = 350 nm).

3. Solvatochromism data for other compounds containing different substituents must be presented in the supplementary information and explained appropriately.

- According to the reviewer's suggestion, a complete set of solvatochromic data of other emissive EliF molecules (i.e., **EliF-2a**, **EliF-2c**, and **EliF-3a**) is now provided as **Supplementary Fig. 9** in the revised SI, and referenced in the main text. Using the newly acquired data, corrections were made for the fluorescence quantum yield, emission wavelength, and the emission spectra of **EliF-2c** in **Fig. 3**, **Supplementary Fig. 11**, **Supplementary Fig. 17**, and **Supplementary Table 1**.
- Consistent with our design principles outlined in **Fig. 1**, borylation of the parent EliF scaffold simplifies both the *conformational space* (by suppressing bond-rotating motions) and *de-excitation pathways* (by eliminating ESIPT), thereby diminishing the solvatochromic response. As a result, **EliF-2a**, **EliF-2c**, and **EliF-3a** all behave similarly: no significant shifts in the fluorescence wavelength with varying solvent polarity and HBD/HBA ability.
- In terms of the fluorescence intensity, however, **EliF-2c** shows the most prominent enhancement when the environment was changed from water to organic solvents (see **Supplementary Fig. 10**). As such, **EliF-2c** remains essentially quenched in aqueous physiological conditions, but responds to even subtle changes in microenvironment upon localization in ER and interaction with ER-enriched proteins. This environment-sensitive emission property makes **EliF-2c** an ideal probe to detect ER stress and avian influenza virus infection.

Supplementary Fig. 9. Solvent polarity-dependent changes in the normalized fluorescence spectra of (a) **EliF-2a**, (b) **EliF-2c**, and (c) **EliF-3a** in cyclohexane, Et₂O, CHCl₃, DMSO, MeCN, iPrOH, EtOH, and water (from front to back; corresponding to increasing solvent polarity as denoted by the E_T(30) values in the y-axis; T = 293 K; A_{exc} = 350 nm).

- . The following text is now added to the revised manuscript to describe the overarching implications of the structure-dependent solvatochromic behavior.

"Replacement of the N–H--N hydrogen bonds with rigid B–N coordinate bonds simplifies both the conformational space and excited-state structural dynamics to produce a brighter emitter, but lowers its sensitivity to the environment." (Page 6, Paragraph 1)

4. The excited state lifetime of the compounds must be presented in the manuscript.

- According to the reviewer's suggestion, the fluorescence lifetimes of EliF molecules were determined by TCSPC technique. The data is now provided in the revised **Supplementary Table 1** (see next page). Information on the instrument used in the TCSPC experiment is also added to the **Physical Measurements** section of the revised manuscript. This additional study sharpened our understanding of the de-excitation mechanisms, for which we are truly grateful to the reviewer.

Supplementary Table 1. Photophysical properties of EliF molecules in CHCl₃ solution ($T = 293$ K).

	$\lambda_{\text{abs,max}}$ (nm)	$\lambda_{\text{em,max}}$ (nm)	ϵ (cm ⁻¹ ·M ⁻¹) ^a	F (%)	τ (ns) ^b	k_r (10 ⁸ s ⁻¹)	k_{nr} (10 ⁸ s ⁻¹)
EliF-1a	345	–	3.9x10 ⁴	< 1			
EliF-1b	350	440 530	4.3x10 ⁴	13	0.22 (-0.110), 1.47 (0.080)	0.50	
EliF-1c	350	–	3.9x10 ⁴	< 1			
EliF-1d	345	410 520	3.7x10 ⁴	18	0.27 (-0.097), 1.83 (0.074)	0.34	
EliF-2a	385	480	1.6x10 ⁴	35	2.11 (0.012), 4.80 (0.037)	0.73 ^c	1.4 ^c
EliF-2b	385	450	1.6x10 ⁴	69	2.41	2.9	1.3
EliF-2c	385	480	1.5x10 ⁴	34	0.61 (0.010), 5.29 (0.042)	0.64 ^c	1.2 ^c
EliF-2d	385	445	1.6x10 ⁴	68	2.21	3.1	1.4
EliF-3a	390	445	3.5x10 ⁴	17	1.03 (0.060), 3.14 (0.004)	1.7 ^c	8.1 ^c
EliF-3b	390	400 425	3.2x10 ⁴	20	0.43 (0.086), 2.47 (0.004)	0.46	4.7 ^c 19 ^c
EliF-3c	390, 470 (broad)	–	3.1x10 ⁴	< 1			
EliF-3d	370	400 425	2.8x10 ⁴	22	0.49 (0.086), 2.12 (0.002)	0.51	4.5 ^c 16 ^c

^aMeasured at the excitation wavelength of 350 nm.

^bNumbers in the parenthesis are pre-exponential factors of the double exponential fitting.

^cDetermined by treating the fluorescence decay as a single-exponential of the major lifetime component.

- Using fluorescence quantum yield (ϕ_f) and lifetime (τ), we were able to determine the rate constants for the radiative (k_r) and non-radiative k_{nr} decay. Compared with monoborylated **EliF-2**, the corresponding diborylated **EliF-3** series molecules have consistently larger k_r values (**Supplementary Table 1**). At the same time, they also suffer from a more pronounced increase in the k_{nr} values. With rapid non-radiative pathways outperforming the radiative de-excitation, **EliF-3** becomes less emissive (i.e., lower F values) than **EliF-2**. This interpretation fully corroborates our previous statement of "... overborylation of **EliF-2** to **EliF-3** . . . opens up undesired non-radiative channels that diminish the fluorescence efficiency", which is now expanded to read:

"overborylation of **EliF-2** to **EliF-3** . . . opens up undesired non-radiative channels that diminish the fluorescence efficiency, as reflected on the increased k_{nr} values determined by lifetime measurements (Supplementary Table 1)" (Page 6, Paragraph 2)

5. HOMO and LUMO of EliF-1b is missing in the supplementary information (Figure S10). Similarly HOMO and LUMO of other compounds (1a, 1c, 1d, 2a, 2c, 2d, etc.) is missing.

- According to the reviewer's suggestion, we computed the frontier molecular orbitals (FMOs) for all of the EliF molecules, and provided them in **Supplementary Figs. 14–16**. For most of the molecules, the HOMO-to-LUMO transition leads to the S_1 excited-state. The only exception is **EliF-3d** undergoing HOMO–1-to-LUMO electronic transition.

Supplementary Fig. 14. Frontier molecular orbitals (FMOs) involved in the transitions to the S_1 excited states of **EliF-1** series examined by TD-DFT computational studies (B3LYP/6-31G(d) level of theory). Numbers in parenthesis are CI (configuration interaction) expansion coefficients, the squared values of which correspond to the contribution of each FMO to the $S_0 - S_1$ transition.

Supplementary Fig. 15. Frontier molecular orbitals (FMOs) involved in the transitions to the S_1 excited states of **EliF-2** series examined by TD-DFT computational studies (B3LYP/6-31G(d) level of theory). Numbers in parenthesis are CI (configuration interaction) expansion coefficients, the squared values of which correspond to the contribution of each FMO to the $S_0 - S_1$ transition.

Supplementary Fig. 16. Frontier molecular orbitals (FMOs) involved in the transitions to the S_1 excited states of **EliF-3** series examined by TD-DFT computational studies (B3LYP/6-31G(d) level of theory). Numbers in parenthesis are CI (configuration interaction) expansion coefficients, the squared values of which correspond to the contribution of each FMO to the $S_0 \rightarrow S_1$ transition.

- For the **EliF-3** series molecules (including **EliF-3d**), all major FMO transitions to the S_1 state entail *vertical-to-horizontal* redistribution of the electron density. Such intramolecular charge transfer (ICT) type transition leads to dark ^1ICT state, which diminishes the emission quantum yields. For **EliF-3c** having an electron-donating dimethylamino group as part of the *vertical* component of the T-shaped π -conjugation, such ICT process becomes even more pronounced, making the molecule essentially non-fluorescent (**Fig. 3**).

6. It is also interesting to see how the HOMO and LUMO is there when diethylamine group is present in the molecule.

- . To answer this question, we constructed DFT computational models **EliF-1c'**, **EliF-2c'**, and **EliF-3c'**, and calculated their HOMO and LUMO. As shown in **Fig. R4** on the next page, the FMOs involved in the electronic transitions are very similar to those of **EliF-1c**, **EliF-2c**, and **EliF-3c** having dimethylamino groups (**Supplementary Figs. 14–15**). Considering that they differ only in the length of the alkyl substituents on the amino group (i.e., $-\text{N}(\text{CH}_3)_2$ vs $-\text{N}(\text{C}_2\text{H}_5)_2$) the results are not unusual.

- With the electron-donating diethylamino group, the HOMO is localized at the *vertical* region of the T-shaped ir-conjugation, whereas the LUMO is localized at the *horizontal* region. A spatial separation of FMOs would thus promote vertical-to-horizontal ICT-type transition (see our response to **Point #5** above).

7. To justify the use of CN derivative in the application the emission properties of other compounds should be decently detailed in the manuscript.

- . As detailed in the manuscript, the choice of the probe molecule for application in AI infection profiling is based on the several key criteria that we systematically imposed on the screening process:
 - (1) Brightness of fluorescence staining:** A total of 12 EliF molecules were initially screened (Page 8, Paragraph 3), to rule out **EliF-1a**, **EliF-1c**, and **EliF-3c**, which showed either no or extremely dim staining pattern (**Supplementary Fig. 20**)
 - (2) Co-localization:** To verify which organelle(s) is labeled by the EliF dyes, co-localization studies were carried out using a series of commercially available organelle trackers (Page 9, Paragraph 1). **EliF-2a**, **EliF-2b**, **EliF-3a**, and **EliF-3b** were eliminated at this stage; they showed fluorescence in all microscope filter sets, and thus could not be examined by co-staining method.
 - (3) Fluorescence response to ER stress conditions:** After verifying that remaining EliF dyes localize in ER (**Supplementary Figs. 21–25**), we evaluated their fluorescence responses to ER stress conditions by flow cytometry-based screening (Page 12, Paragraph 4). Only **EliF-1b** and **EliF-2c** showed fluorescence enhancement with increasing dose of ER stress-inducing agents (**Fig. 7**).
 - (4) Applications in AI infection profiling:** With largest fluorescence enhancement to environmental change (see our response to **Point #3** above), stronger fluorescence response

under ER stress conditions (**Fig. 7**), and lower photo-cytotoxicity (see our response to **Reviewer #1's Point #4** above), **EliF-2c** outperforms **EliF-1b**. Our AI infection profiling studies were thus carried out with **EliF-2c**.

. To make this point clear, the following sentence was added in the revised manuscript:

"Among the final two candidates, **EliF-2c** was employed as the probe because of its stronger fluorescence response than **EliF-1b** under ER stress conditions (**Fig. 7**); an added bonus was its lower photo-cytotoxicity (**Supplementary Fig. 19**)."
(Page 14, Paragraph 3)

Reviewer #3. We thank the reviewer for a thorough inspection and critical comments on the crystallographic structure analysis and DFT computational studies. Listed below are our point-by-point responses.

1. One structure (elif-et_1) has an oxygen in the formula where there isn't one. In many of the structure, the Z' needs to be updated prior to updating the moiety and sum formulas.

. We thank the reviewer for bringing these oversights to our attention. Corrections were made for consistency in: chemical formula of **EliF-2a** (datablock: elif-et_1); Z value of **EliF-2b** (datablock: elif-cn_1); value of **EliF-2d** (datablock: exp62). The revised CIF files and corresponding checkCIF files are included in this revised submission.

. The crystallographic data deposited at the Cambridge Crystallographic Data Centre (CCDC) reflects these changes: 2003263 for **EliF-2a**, 2003267 for **EliF-2b**, and 2003268 for **EliF-2d**.

2. The SC-XRD instruments used should also be listed in the "Physical Measurements" section.

. According to the reviewer's suggestion, the **Physical Measurements** section of the manuscript has been revised to include information on the instrument used in the single-crystal X-ray crystallographic studies. The added text reads "Single-crystal X-ray diffraction studies were carried out using Cu K α radiation on an XtaLAB AFC12 (RINC): Kappa dual home/near diffractometer or a SuperNova, Dual, Cu at zero, AtlasS2 diffractometer."

0. For the computed structures, are they a global (or local) minimum or first-order saddle points (transition states) or higher (meaningless)? Were frequency calculations performed to verify this? If not, please do so, and report the Cartesian coordinates of the computed structure in the supporting information.

. The computed DFT models of **EliF-2b** and **EliF-3b** had already been confirmed as true minima with no imaginary frequencies found in the vibrational frequency calculations with Gaussian 09.

- . The validity of our previous calculations was further checked by using Gaussian 16, the latest in the Gaussian series of programs having a finer integration grid than the earlier version (Gaussian 09). For consistency, we also double-checked the FMOs and electronic excitations of other EliF molecules with Gaussian 16 to arrive at the same conclusion. The **Computational Studies** section of the revised manuscript makes these points clear.
- . According to the reviewer's suggestion, the Cartesian coordinates of all of the computed structures are provided in **Supplementary Tables 5–19** of the revised SI.

REVIEWERS' COMMENTS

Reviewer #1 (Remarks to the Author):

The authors revised the manuscript carefully and answer all the comments from the reviewers. This work can be accepted at the current version.

Reviewer #3 (Remarks to the Author):

The authors have addressed my comments sufficiently. I can recommend acceptance of this manuscript.

Reviewer #4 (Remarks to the Author):

The authors have solved all the problems raised, and the scientific quality of this work has been improved. Thus, this manuscript can be accepted for publication in its current form.

Re: Ms# NCOMMS-21-45703A "Orthogonally-Tunable and ER-Targeting Fluorophores Detect Avian Influenza Virus Early Infection" by Kang, Haque, Lee, Hong, Hong, Kim, Lee, Lee, and Lee.

Reviewer #1 (Remarks to the Author): *"The authors revised the manuscript carefully and answer all the comments from the reviewers. This work can be accepted at the current version."*

- ▶ Reviewer #1 recommended accepting the work "at the current version". Therefore, we made no changes in the scientific or technical aspects of the current manuscript.

Reviewer #3 (Remarks to the Author): *"The authors have addressed my comments sufficiently. I can recommend acceptance of this manuscript."*

- ▶ Reviewer #3 recommended "acceptance of this manuscript". Therefore, we made no changes in the scientific or technical aspects of the current version.

Reviewer #4 (Remarks to the Author): *"The authors have solved all the problems raised, and the scientific quality of this work has been improved. Thus, this manuscript can be accepted for publication in its current form."*

- ▶ Reviewer #4 recommended "publication in its current form". Therefore, we made no changes in the scientific or technical aspects of the current version.
-